# Concomitant Calcium Channelopathies Involving *CACNA1A* and *CACNA1F:* A Case Report and Review of the Literature

**DOI:** 10.3390/genes14020400

**Published:** 2023-02-03

**Authors:** Donna Schaare, Sara M. Sarasua, Laina Lusk, Shridhar Parthasarathy, Liangjiang Wang, Ingo Helbig, Luigi Boccuto

**Affiliations:** 1Ph.D. Program in Healthcare Genetics and Genomics, School of Nursing, College of Behavioral, Social and Health Sciences, Clemson University, Clemson, SC 29634, USA; 2Division of Neurology, Children’s Hospital of Philadelphia, Philadelphia, PA 19104, USA; 3Department of Genetics and Biochemistry, College of Science, Clemson University, Clemson, SC 29634, USA; 4Department of Neurology, Perelman School of Medicine, University of Pennsylvania, Philadelphia, PA 19104, USA

**Keywords:** *CACNA1A*, *CACNA1F*, channelopathy, calcium channel, hemiplegic migraine, sporadic hemiplegic migraine type 1 (SHM1), immune dysfunction

## Abstract

Calcium channels are an integral component in maintaining cellular function. Alterations may lead to channelopathies, primarily manifested in the central nervous system. This study describes the clinical and genetic features of a unique 12-year-old boy harboring two congenital calcium channelopathies, involving the *CACNA1A* and *CACNA1F* genes, and provides an unadulterated view of the natural history of sporadic hemiplegic migraine type 1 (SHM1) due to the patient’s inability to tolerate any preventative medication. The patient presents with episodes of vomiting, hemiplegia, cerebral edema, seizure, fever, transient blindness, and encephalopathy. He is nonverbal, nonambulatory, and forced to have a very limited diet due to abnormal immune responses. The SHM1 manifestations apparent in the subject are consistent with the phenotype described in the 48 patients identified as part of a systematic literature review. The ocular symptoms of *CACNA1F* align with the family history of the subject. The presence of multiple pathogenic variants make it difficult to identify a clear phenotype–genotype correlation in the present case. Moreover, the detailed case description and natural history along with the comprehensive review of the literature contribute to the understanding of this complex disorder and point to the need for comprehensive clinical assessments of SHM1.

## 1. Introduction

### 1.1. Voltage-Gated Calcium Channels

Excitable cells are defined as containing voltage-gated ion channels and responding to depolarization by triggering electrical impulses. Voltage-gated calcium channels (VGCCs) rely on calcium (Ca^2+^) to induce the change in the action potential. They function as an integral part of cellular regulation, both in maintaining membrane potential as well as controlling intracellular Ca^2+^ levels [1,2]. Due to the calcium channel’s key role in cell activities, any alteration to their structure will modify gating, resulting in too much or too little Ca^2+^ intracellularly [1,2]. Not only does the channel change alter the cell’s ability to respond to depolarization, but also its ability to regulate essential processes such as gene transcription, neurotransmitter release, hormone secretion, and enzymatic activity, resulting in multisystem dysregulation and diseases, termed “calcium channelopathies” [1,2]. The VGCC superfamily includes ten genes (Table 1) with tissue-dependent expression patterns [2]. The L-type, P/Q-type, N-type, R-type, and T-type channels have all been shown to be expressed in the brain as well as in other tissues to varying degrees [1,2]. An example of the pervasive effects of calcium channelopathies is apparent in patients carrying mutations in *CACNA1C,* which are associated with Timothy syndrome, a multiorgan condition causing cardiovascular issues, cognitive impairment, autism, periodic hypoglycemia, and immune deficits [1].

### 1.2. SHM1 a CACNA1A-Related Phenotype

Hemiplegic migraine is an atypical acute form of migraine currently known to be caused by genetic mutations in *CACNA1A*, *ATP1A2*, *SCN1A*, and *PRRT2* [3]. As an allelic disease, *CACNA1A* pathogenic variants result in multiple phenotypes, including hemiplegic migraine, ataxia, and epilepsy [3,4,5]. The hemiplegic migraine produced by pathogenic *CACNA1A* variants can be as mild as transient episodes of unilateral paralysis with headache or as severe as long-term hemiparesis with coma, seizure, respiratory distress, cerebral edema, fever, acute encephalopathy, and ataxia [3,4,6,7,8]. Although the hereditary form of hemiplegic migraine caused by *CACNA1A* variants—familial hemiplegic migraine 1, or FHM1 (OMIM #141500)—has been studied extensively, the far broader and more severe sporadic form of the disorder that is produced by de novo variants—sporadic hemiplegic migraine 1, or SHM1—has had limited clinical description in humans [5,8].

*CACNA1A* maps on chromosome 19p13.1 and is translated into the main protein, subunit α 1A of the P/Q type voltage-gated neuronal channel, Ca_V_2.1. The *CACNA1A* gene product contains six membrane-spanning regions (S1–S6) that comprise four equivalent domains (DI-DIV). The S4 region acts as a voltage sensor, while S5-loop-S6 is the Ca^2+^-discerning pore. The channel facilitates Ca^2+^ ion entry into the cell contributing to the maintenance of calcium homeostasis, which is integral in multiple cellular pathways, including gene expression and neurotransmitter discharge [9,10]. Alterations to the channel can cause either loss of function (LOF), in which channel production or activity is significantly reduced, or gain of function (GOF), in which the channels are produced but have abnormal properties. LOF variants have been linked to episodic ataxia and epilepsy [10]. GOF variants have similarly been associated with ataxia, developmental delay, and epilepsy, but also hemiplegic migraine. It has been postulated that the GOF variants associated with hemiplegic migraine results in increased susceptibility to neuronal hyperexcitability with an abnormal glutamate release and decreased ceiling for cortical spreading depression, which is the initiation of hemiplegic migraine [11,12].

Investigators hypothesize that the type of alteration, GOF or LOF, as well as variant location within the gene result in the spectrums of phenotypic presentations [4,5,11]. Due to small cohorts and short-term follow-up, these correlations have eluded current investigations [5].

### 1.3. CACNA1F-Related Phenotype

*CACNA1F* is located on chromosome Xp11.23, comprising 48 exons, and produces a subunit of the L-type calcium channel Ca_v_1.4. Ca_v_1.4 is structurally similar to Cav2.1, with six transmembrane regions (S1–S6), repeated in four equivalent domains (I–IV) [13]. However, as an L-type channel, it is responsible for the transfer of large amounts of current [14]. Initially, Ca_v_1.4 was believed to be expressed only in the retina. However, it was more recently discovered in the adrenal glands, bone marrow, brain, muscle, spine, spleen, and thymus, with implications regarding potential immune regulation [13,14].

In retinal neurons, Ca_v_1.4 channels are situated at photoreceptor terminals and are responsible for the tonic Ca^2+^ entry needed to facilitate a sustained neurotransmitter release [13]. Without the channel’s unique properties, including activation at a negative potential and delayed voltage-dependent deactivation, the retina would not function properly [14,15]. Alterations in the Ca_v_1.4 structure have been shown to result in a host of retinal disorders, including incomplete congenital stationary night blindness type 2 (CSNB2, OMIM #3000071), Aland island eye disease (AIED, OMIM #300600), and X-linked rod and cone dystrophy type 3 (CORDX3, OMIM #300476) [14,16,17].

With the identification of Ca_v_1.4 in immune system tissue, including bone, spleen, lymph nodes, and thymus, researchers investigated how a voltage-gated channel may impact a non-excitable cell, a point lacking current consensus [14,18]. Over the next few years, multiple scientists uncovered that L-type calcium channels, specifically Ca_v_1.4, were expressed in lymphocytes and postulated the integral role in T lymphocyte development, regulation, activation, and death through optimal Ca^2+^ signaling [18,19,20,21]. Additionally uncovered were novel splice variants, Ca_v_1.4a and Ca_v_1.4b, that alter the voltage-gating properties of the channel [22]. In Ca_v_1.4a, exons 31–34 and 37 are removed, which eliminates the voltage sensor for domain IV [17,22]. Alternative splicing in Ca_v_1.4b removes exons affecting the potential voltage gating properties and causes a frameshift resulting in 40% homology with Ca_v_1.1 at the C-terminus [22]. This channel variant is no longer responsive to membrane depolarization and potentially responsive to antigen binding instead [18]. Studies have demonstrated that Ca_v_1.4 deficient mice express a phenotype of T lymphocyte dysfunction and potentially immunodeficiency [21]. The impact of the *CACNA1F* gene on the immune system is highly controversial and still under debate.

### 1.4. Case Report

This case report describes the first patient ever documented with two concomitant pathogenic variants in genes coding for Ca^2+^ channel proteins, *CACNA1A* and *CACNA1F*. Additionally, this patient provides a “pure” natural history of *CACNA1A*-related SHM1 due to his incapacity to tolerate any oral therapeutics, a valuable contribution to generalizable knowledge about the disorder. The resulting clinical presentation is complex, possibly due to multiple phenotypes and the systemic effects of Ca^2+^ channelopathies. SHM1 has been coupled with a diverse group of signs, and the complete phenotype may include other established disorders. Therefore, the purpose of this case report is to document the longitudinal history of SHM1 and compare this case to the phenotypes from pathogenic variants noted in the literature to elucidate where the SHM1 phenotype ends, and the *CACNA1F* phenotype begins. With most published data focused on the milder familial form (FHM1), there is a misconception that headache with hemiparesis is the only manifestation in a patient with SHM1 when seen by clinicians (Appendix A). This fallacy can lead to unnecessary invasive testing, improper therapeutic interventions, and delays in obtaining an accurate diagnosis [23,24,25]. Regarding the second identified pathogenic variant, research on the *CACNA1F* gene encoding the L-type channel has primarily focused on its manifestations in the eyes, with recent studies highlighting its expression in other tissues [14,18]. Through the description of the longitudinal disease history presented by the reported case, this study strives to broaden the clinicians’ knowledge on diagnosing and managing complex patients with simultaneous rare disorders.

### 1.5. Objectives

This investigation aimed to present a unique case study of a child harboring two calcium channel pathogenic variants, *CACNA1A* and *CACNA1F,* and to describe a clear natural history of the SHM1 disorder, never illustrated in the literature before due to the complication of concomitant therapeutics. An additional aim of this study was to conduct an original systematic literature review to describe the complete phenotype of SHM1 related to *CACNA1A* variants both during hemiplegic migraine episodes and outside of attacks, since all previous reviews have focused on the milder familial form (FHM1). Finally, the case was compared to the complete SHM1 phenotype described in the review to determine where the *CACNA1F* variant may be influencing the expressed phenotype in the patient.

## 2. Methods

### 2.1. Ethical Protocols

The case study protocol complies with the principles of the Declaration of Helsinki and was approved by the Institutional Review Board at Children’s Hospital of Philadelphia. Written informed consent was obtained from the subject’s parents in the case report prior to the initiation of the study.

### 2.2. Systematic Review Procedure

The Preferred Reporting Items for Systematic Reviews and Meta-Analyses (PRISMA) [26] guidelines were utilized to perform a systematic literature review. The search strategy is included in Section 2.3, and the inclusion/exclusion criteria are contained in Section 2.4. A two-step process was employed to screen the original article pool, beginning with the titles/abstracts, and followed by the entire text. Zotero Citation Manager (Version 5.0.96.3; Roy Rosenzweig Center for History and New Media, 2021) and the PRISMA flow diagram map captured the articles and the sifting process, respectively (Appendix A). Pertinent data were abstracted from reviewed publications and compiled.

### 2.3. Search Strategy

MEDLINE, through EBSCO, PubMed via NIH, and Web of Science by way of Clarivate, were the databases searched. The terminology was developed in consultation with a research librarian employing keywords and MeSH terms (Appendix A). To ensure all patients were captured, the term “Sporadic” was not included, since de novo patients are often incorrectly diagnosed with familial hemiplegic migraine type 1 (FHM1), not sporadic hemiplegic migraine type 1 (SHM1). Additional articles were identified through bibliography searches with matching inclusion criteria.

### 2.4. Criteria for Inclusion

Inclusion criteria included published manuscripts in peer-reviewed journals. Dissertations, meetings, congresses, or any other gray literature were omitted. Only articles published in English were integrated. Since *CACNA1A* was first identified in 1996, the included publication dates spanned from 1996 to 11 May 2022, when the search was terminated. Literature titles and abstracts were reviewed for phenotypic information on human patients diagnosed with hemiplegic migraine from a de novo *CACNA1A* variant. Articles describing pathophysiology, functional studies, or therapeutics lacking phenotypic information were excluded. Studies centered on animals, or cell culture but not on human patients were excluded. After the title and abstract review, all included articles were subject to full-text analysis.

### 2.5. Data Collection

The primary researcher abstracted the information from the studies during the complete article analysis. An excel spreadsheet was crafted for data collection with column titles including study design, authors, publication year, location, number of patients, patient sex, age at diagnosis, variant, phenotype reported information, additional clinical manifestations, unique findings, limitations, and article citation in APA format (Appendix A). To better summarize the symptoms experienced during a hemiplegic attack, another table was created in excel describing each participant included in the first review (Appendix A). All the phenotypic presentations outside of hemiplegic episodes were extracted from the original table and transferred to a new chart that links the participant to the features (Appendix A).

## 3. Case Report

A 12-year-old Caucasian male without any dysmorphic features was diagnosed at seven years of age with SHM1. At the evaluation, his height was 145 cm (30th centile), his weight was 39 kg (46th centile), and his head circumference was 55.5 cm (75th centile). The subject suffers periodic attacks that include migraine with hemiparesis in conjunction with noninfectious fever, vomiting, encephalopathy, seizure, cerebral edema, and transient blindness. Outside of attacks, the patient exhibits hypotonia, and paroxysmal events, including dyskinesia, and has documented cerebellar atrophy since 2014 at age 4. The patient experiences abnormal responses of the immune system to the exposure to all food outside of pork, sweet potatoes, and apple juice. If this extremely limited diet is altered, the result is a severe hemiplegic attack with cerebral edema and status epilepticus. He is nonverbal and nonambulatory and has been diagnosed with autism spectrum disorder and intellectual disability. Currently, the patient is not on any preventative therapeutic interventions.

### 3.1. Clinical Findings

#### 3.1.1. Developmental History

The patient is the youngest child in a family with two older healthy siblings, a maternal half-sister, and a full sister. The parents suffered three miscarriages—including one case with trisomy 22—before the patient’s birth. His maternal uncle died at age 17 from complications from coarctation of the aorta. He was exposed, in utero, to pityriasis rosea during the 16th week of gestation. Born at 39 weeks of gestation via vaginal birth after induction due to complications from preeclampsia and immune thrombocytopenic purpura (ITP), the patient had a posterior occipital presentation. At birth, his weight was 3.912 kg (75th centile), and he was 50.8 cm long (75th centile), with a 34 cm head circumference (50th centile), and his Apgar score was 9 both at 1 min and at 5 min. The patient experienced developmental delay. He rolled at one year old but could not sit without support at 15 months. Due to the severity of the developmental delay, the patient received physical therapy as well as occupational and speech therapy with slow improvement. By the age of five years, he could sit independently and crawl on his abdomen to move around the room. Since the age of seven, with assistance and the use of a walker, the patient can ambulate for short durations.

#### 3.1.2. Neurological History

By three months of age, the child was seen by pediatric neurology and diagnosed with moderate hypotonia, mild nonfixed torticollis, mild inconsistent “sun-setting” or downward-gaze, and megacephaly (>97° centile). At 21 months, the patient suffered his first attack. Figure 1 provides a graphic of the natural history of the disorder over time. Table 2 offers a synopsis of attacks (see Appendix A for a more detailed description of each attack). He suffered from episodes of transient hemiplegia and vomiting, which became more frequent at 11 years of age and included paroxysmal dyskinesia. At that time, a propensity for focal seizures due to spikes seen on EEGs during hemiplegic attacks was identified. It was later determined that the patient was inadvertently exposed to mold in the home for over a year during an increased frequency of hemiplegic events. It is unclear why the presentation of the more recent episodes, post-puberty, lacked coma, lethargy, and vomiting and included involuntary movements with impaired consciousness. It may be due to the evolution of the disorder or tied to medications used to treat each attack.

#### 3.1.3. Medical History

Within his first year, the patient developed periodic noninfectious fevers (102–103° F) for 24–48 h with and without hives that continued throughout his first decade. After the first “attack”, an allergist/immunologist evaluated the patient, who suspected Food protein-induced enterocolitis syndrome (FPIES). Radioallergosorbent test (RAST) testing uncovered a severe airborne allergic reaction to peanuts and multiple other IgE-mediated allergens. The allergist recommended removing all foods from the patient’s diet and reintroducing them one at a time to determine intolerances, which resulted in identification of toleration of only the two foods the patient tolerated and is currently eating. All subsequent food trials resulted in moderate to severe episodes. Due to his immune issues, the patient is homeschooled to reduce the chances of exposure to airborne allergens that have been linked to past episodes. The patient was diagnosed with scoliosis and hip dysplasia, which were addressed through physical therapy and monitored for progression by an orthopedist. At six years of age, the patient was diagnosed with myopia, hyperopia, astigmatism, and “abnormal” gaze development.

### 3.2. Diagnostic Assessment

Multiple diagnostic testing modalities were employed, including basic metabolic and genetic assays, general laboratory panels, neurologic examination through magnetic resonance imaging (MRI), electroencephalogram (EEG), autism evaluation, cardiac studies, ocular evaluations, and multiple immune panels.

#### 3.2.1. Genetic and Metabolic Testing

Initial genetic and metabolic testing were all negative and included a newborn screening panel, urine organic acids, blood amino acids, chromosomal microarray, and mitochondrial testing. After the diagnosis of SHM1, the family traveled to see a specialist who recommended whole-exome sequencing, suspecting multiple pathogenic variants. The exome was performed as a trio (patient and both parents) by GeneDx and revealed two likely pathogenic variants in the proband: a de novo missense c.2102G>A, p.Gly701Glu variant in *CACNA1A*, and a maternally inherited c.2322-2G>A variant in *CACNA1F*, disrupting the canonical splice acceptor site in intron 17. The mother of the patient is affected with congenital stationary night blindness, a *CACNA1F*-associated disorder. The full sister also has the familial *CACNA1F* variant but is unaffected. It is unknown whether the truncated CACNA1F protein results in an abnormal protein or nonsense-mediated mRNA decay. A third pathogenic variant in the *BRCA2* gene was identified (c.4808dupA, p.N1603KfsX6) but has no impact on the patient’s neurological health. Since this variant is associated with an increased risk of multiple cancers in adulthood, future screenings have been recommended.

#### 3.2.2. MRI

An initial MRI was performed during the first severe attack at age 21 months, and it was suspicious for encephalitis. Subsequent MRIs were performed at every severe HM event, one outside of the HM event, and rapid sequences or abbreviated MRIs at minor HM events (Figure 2).

#### 3.2.3. EEGs

EEGs were captured on all severe attacks and were initially normal until the onset of hemiplegia during the third attack. The one-sided paralysis was apparent in slowing on the corresponding side, with a slow background pattern observed throughout all subsequent EEGs. The slowing was seen when hemiplegia occurred. Diffuse encephalopathy with the right hemisphere demonstrating more cortical dysfunction as well as focal spikes were noted during his last EEG at 11 years and 9 months.

#### 3.2.4. Labs

The patient’s labs during an episode always suggested sepsis, including high WBC and CRP, yet the cultures verified the noninfectious nature of the symptoms. CSF findings documented lymphocytic pleocytosis on numerous occasions. Labs consistently revealed acidosis, high glucose, and low calcium.

#### 3.2.5. Cardiac Evaluation

Due to a family history of congenital cardiac abnormalities, the patient was evaluated by a pediatric cardiologist. An echocardiogram and EKG were performed at three years of age and again at age seven years eight months during a severe attack when a murmur was noted with unremarkable findings.

#### 3.2.6. Ocular Assessment

The patient was initially evaluated by a pediatric ophthalmologist at two years of age and again after losing his vision during his fifth severe attack. Ophthalmological findings were unremarkable, with a normal optic nerve. Within three months, the patient’s vision returned. Subsequently, it was determined by an ophthalmologist that cortical blindness was the cause of his temporary loss of vision post-attacks.

#### 3.2.7. Immune Analysis

Initial testing at age 2 looked at immune markers revealing consistently high IFN-g. An autoimmune panel was negative. At age 4, an investigation into the patient’s T and B Cells revealed low CD3 1382 (T Cells) (normal range 1484–5327 cells/µL), and an immunoglobulin assay uncovered a low IgG score 642 (normal range 700–1600 mg/dL). Once diagnosed with *CACNA1F*, an immunologist ran an extensive immune panel, including the proinflammatory cytokine panel, extended lymphocyte panel, B Cell panel, functional natural killer assay, sedimentation rate, and C-reactive protein combo, with unremarkable findings. This was performed during a time of no attacks and showed high IFN-g 8.3 pg/mL (ref range 0.0–6.5 pg/mL), low CD19^+^/CD27^+^/IgD^+^ 2.4% (ref range 3.2–16.2%), low CD19^+^/CD27^+^/IgD^−^ 3.5% (ref range 4.6–33.7%), low CD19^+^/CD27^+^/IgM^+^ 2.9% (ref range 3.8–18.6%), low CD19^+^/CD27^+^/IgM^−^ 3.0% (ref range 3.9–30.7%), and high CD5^+^/CD19^+^ 6.5% (ref range 0.0–5.2%). All other values were normal, including the sed rate and CRP.

### 3.3. Therapeutic Interventions and Follow-Up

Due to an inability to tolerate most foods and a hyperimmune response, all daily preventative therapeutic interventions have failed. Within a few hours of starting verapamil 40 mg at age seven, the patient developed transient premature ventricular contractions and accelerated ventricular rhythm, although the reaction cleared as was seen on a 24 h Holter monitor. Within a few days, the patient vomited shortly after taking the medication, and it was discontinued. During the most recent attack, the protocol by Camia et al. [27], which consisted of dexamethasone IV 0.5 mg/kg/day, three pulses for up to three days, was employed and resulted in a significant reduction in the duration and severity of the attack.

## 4. A Review of the Literature

### 4.1. SHM1 a CACNA1A-Related Phenotype

#### 4.1.1. Demographic Information, Variant Location, and Variant Substitution

A total of 48 patients with SHM1 caused by de novo *CACNA1A* variants were identified in 28 articles in this systematic literature review (Appendix A) [3,4,5,6,7,8,10,12,23,24,25,27,28,29,30,31,32,33,34,35,36,37,38,39,40,41]. The population and reported ages are included in Appendix A.

Twenty variants were identified among the 48 patients: 18 of them were missense, one was a deletion, and one disrupted a splice site. Almost all the variants were located either in the voltage sensor (S4) or in the pore-forming channel (S5-loop-S6) of any of the four domains (DI-DIV). The two exceptions were one alteration of a splice site (c.3825+1G>A) and the p.Phe363Ser substitution (p.F363S in Figure 3), which is situated on the I–II linker near the α-interaction domain (AID). The highest concentration of mutations, 9 of the 20, were positioned in domain III (45.0%), and 4 of them spanned the S4 voltage sensor (Figure 3). The amino acid substitutions varied from staying within polarity and hydrophilic/hydrophobic grouping to significant alterations such as going from nonpolar to polar or vice versa. Moreover, some shifts included exchanging a positively charged amino acid for a neutral one.

#### 4.1.2. Hemiplegic Migraine Presentation during Attacks

The symptoms observed during a hemiplegic attack in patients with *CACNA1A* variants were described in 33 of the 48 patients. Hemiplegia and headache were apparent in all patients. Some of the most common additional symptoms included coma/altered consciousness (75.8%), seizure including status epilepticus (75.8%), cerebral edema (45.5%), noninfectious fever (51.5%), and vomiting/nausea (39.4%) (full list provided in Table 3). The age of the first presentation for hemiplegic migraine ranged between 18 months [6,8] and 28 years [25]. The common triggers that were noted included minor head trauma (45.5%), fatigue/sleep deprivation (9.1%), sun exposure (6.1%), and stress (6.1%). Generally, triggers were not recorded or of unknown origin. The duration of attacks could be as short as a few hours or as long as a few months. Most occurrences that followed head trauma occurred at least 30 min after the event, with the level of consciousness declining until the patient no longer responded, becoming comatose. Nearly all head trauma was mild, including falls with no immediate loss of consciousness or minor bumps [4,6,7,8,28,29].

#### 4.1.3. Common Symptoms Outside of Attacks

Symptoms reported outside of hemiplegic attacks included movement disorders (80.4%), intellectual disability (65.2%), global developmental delay (63.0%), brain atrophy (75.7%), abnormal eye movements (47.8%), hypotonia (52.2%), and seizures (26.1%) (full list provided in Table 4). Six patients had no other symptoms other than hemiplegic migraine episodes [25,28,35,41], and two patients’ phenotypes, other than hemiplegic incidents, were not defined (Appendix A) [3,7].

#### 4.1.4. Ataxia, Extra Pyramidal Movement Disorders, and Spasticity

The most common comorbid feature in patients with SHM1 is movement disorders (80.4%). The symptoms range from jitteriness in infancy to progressive ataxia. Ataxia in multiple forms was documented, including early-onset ataxia, which presents within the first twenty-four months of life [10]. One patient (#29) lost the ability to ambulate by age 13 due to progressive cerebellar ataxia [8]. LeRoux et al. expanded on the ataxic phenotype to include other movement disorders, such as dystonia and choreoathetosis [33]. Falling under the dystonia umbrella is hypotonia, which was reported in 52.2% of patients, often occurring in infancy. Pyramidal signs, which include spasticity, weakness, hyperreflexia, and positive Babinski sign, were also noted in this patient population [33]. The tremor was exhibited in 23.9% of patients, leading to a Parkinson-like clinical presentation. Lastly, dyskinesia, an all-inclusive term to represent uncoordinated movements, can be seen periodically in this group (Table 4).

#### 4.1.5. Global Developmental Delay and Intellectual Disability

Global developmental delay (GDD) and intellectual disability (ID) are also highly prevalent in the SHM1 population (Table 4). Delays in head control, sitting, crawling, and walking are widespread. Sometimes, the motor delay resolves, while improvement is not apparent in other cases [5]. Decline in function was observed in one case, where the patient lost the ability to ambulate at an undisclosed age prior to the age of six, after a prolonged hemiplegic event [33]. Included are multiple cases of nonambulatory patients [10,32]. Dysarthria is frequently noted, regarding verbalization and verbal acuity (28.3%). Like lack of ambulation, multiple nonverbal patients are reported throughout this review [30,32,33]. ID is more ubiquitous than GDD and can range from mild to profound. With mild classification correlating to IQ scores from 50 to 69, moderate from 36 to 49, severe from 20 to 35, and profound below 20 [42]. Of the 30 patients identified with ID, 10/30 (33.3%) were not defined, 9/30 (30%) were listed as mild, 4/30 (13.3%) were classified as moderate, and 7/30 (23.4%) were severe with no patients grouped in the profound class [5,6,28,29,31]. For nonverbal patients, the specific degree of impairment can be difficult to assess but estimates using IQ testing and other means were included in many of the studies [5,6,9,10,23,24,28,29,31,33,34,36,39]. Autism spectrum disorder (ASD) was documented in only 3 of the 48 patients (6.25%) [6,33].

#### 4.1.6. Brain Atrophy and Abnormal EEGs

Cerebellar Atrophy is a frequently reported manifestation of SHM1 (75.7%), as seen in MRI and computed tomography (CT). It is usually not an initial finding but typically occurs after multiple hemiplegic attacks later in childhood or early adolescence [4,6,30]. Nearly one-third (32.1%) of cases were reimaged after multiple years, and attacks exhibited progressive atrophy or degeneration (Table 4). Abnormal EEGs are common for patients with SHM1 during hemiplegic attacks (Table 3). The results cited in the literature discuss EEGs measured during hemiplegic attacks and demonstrate slowing backgrounds corresponding to the side of the brain edema and the opposite side of paralysis. Occasionally, spike waves were observed. No baseline nor follow-up post-resolution of hemiplegia EEGs were disclosed (Appendix A) [5,8,12,23,25,31,32,35,39,41].

#### 4.1.7. Paroxysmal Events and Abnormal Eye Movements

Paroxysmal movements consist of periodic unintentional actions and have been recently documented in SHM1 patients before they exhibit hemiplegic attacks. Episodic events may be linked to migraine because both are repeated short-lived assaults [30]. Paroxysmal disorders specific to this patient population include benign paroxysmal torticollis (BPT), paroxysmal tonic upgaze (PTU), benign paroxysmal vertigo (BPV), and paroxysmal ataxia (PA). Nine patients were acknowledged to have at least one of these conditions, and three were diagnosed in infancy (Table 4) [5,6,8,30,34]. Another early signal of SHM1 is abnormal eye movements, specifically nystagmus and strabismus. Nearly 50% of patients in this review presented with either or both features, and many manifested in early childhood (Table 4) [4,5,6,10,24,25,31,32,33,34,40].

#### 4.1.8. Seizures

Seizures in SHM1 can occur during and outside the setting of hemiplegic attacks. Seizures, outside of hemiplegic attacks, occurred in 24% (11/46) of SHM1 patients. They were heterogeneous in type and included tonic–clonic, absence, focal, atonic, and generalized (Table 4) [5,10,24,28,29,32,37,39]. Many more—76% (25/33)—experienced seizures in combination with cerebral edema during hemiplegic attacks. During attacks, seizure types included tonic–clonic, tonic contracture, and hemiclonic (Table 3) [3,4,5,6,7,8,12,23,24,27,28,29,31,32,36,37,38,39]. Status epilepticus was documented in five patients—9% (3/33) during hemiplegic attacks and 4% (2/46) outside of attacks [5,27,28,35].

#### 4.1.9. Other Findings Outside of Hemiplegic Attacks

Less common features included autonomic signs such as tachycardia, low blood pressure, and apnea. Also reported were GI signs such as constipation and abdominal pain; migraine with and without aura; stroke; paralysis episodes; and hallucinations (Appendix A) [6,12,25,27,32,36,37].

### 4.2. A Scoping Review of CACNA1F-Related Phenotypes

#### 4.2.1. Demographic Information, Variant Type, and Variant Location

This scoping literature review identified 54 patients in fifteen journal publications with confirmed *CACNA1F* pathogenic variants (Appendix A) [15,16,43,44,45,46,47,48,49,50,51,52,53,54,55]. The population consisted of 47 males (87.0%) and 7 females (13.0%), consistent with being an X-linked recessive disorder.

Within the 54 patients, thirty-one variants were detected. Twelve missense mutations made up most of the pathogenic variants, followed by eight nonsense mutations, six splice site variants, and five deletions. The variant type and location did not appear to correlate to the severity of the phenotypes captured in this review (Appendix A). Patients with early termination nonsense mutations or deletions did not present as any more severe than those who suffered from later missense variants. Unlike *CACNA1A*, where variants in the voltage sensor and pore-forming channel proved the most damaging, the variants appear equally deleterious throughout the channel. This heterogeneity may be related to the discovery of 19 alternative splice variants during an extensive investigation of the retina, which suggests that different splicing may relate to the mechanism of how calcium influx is maintained in L-type channels, specifically Ca_v_1.4 [18].

#### 4.2.2. Retinal Dystrophy Associated with CACNA1F

The central retinal dystrophies associated with *CACNA1F* are incomplete congenital stationary night blindness (CSNB2A), Aland island eye disease (AIED), and X-linked rod and cone dystrophy type 3 (CORDX3). Of the 54 patients identified in this review, 47 (87.0%) were diagnosed with a form of retinal dystrophy [15,16,43,44,45,46,47,48,49,50,51,52,53,54,55]. CSNB2A was the most prevalent, with 40 out of 47 patients (85.1%) receiving this diagnosis, including 2 patients whose clinical descriptions fit either CSNB2A or AIED [15,53]. Less typical was AIED, with only one patient (2.1%) meeting the criteria. However, as noted by Mahmood and colleagues, the two disorders overlapping on almost every essential feature can make the diagnosis difficult [15]. CORDX3 was also only detected in one patient (2.1%) [55]. Although not listed in OMIM as a *CACNA1F*-associated retinal disorder, retinitis pigmentosa (RP) is affiliated with multiple X-linked genes (RP2-OMIM #312600, RP3-OMIM #300029, RP6-OMIM #312612) and was identified in four patients (8.5%) [54,55]. One other patient (2.1%), highlighted for neurocognitive symptoms, was reported to have Leber’s congenital amaurosis (LCA) as well [55].

#### 4.2.3. Other Ocular Manifestations

Myopia and Nystagmus are two of the key symptoms seen in CSNB2A and AIED. Thirty-three of the 54 patients (61.1%) had some form of nystagmus, and 11 patients (20.4%) were reported to have myopia [15,16,43,44,45,46,47,49,51,52,53,54,55]. By including the diagnosis of CSNB2A or AIED, it is possible that myopia and nystagmus were assumed to be present and not additionally disclosed. Other ocular manifestations included optic atrophy 1/54 (1.9%), cortical blindness 1/54 (1.9%), and nyctalopia 1/54 (1.9%) in a patient without CSNB2A [53,55].

#### 4.2.4. Neurocognitive Signs

With *CACNA1F* expression in the presynaptic regions of the hippocampus and cerebellum and the Ca_v_1.4 channel’s role in neurotransmitter release, alterations in the protein are linked to multiple neurocognitive disorders [55]. Like other channelopathies, intellectual disability is evident in this patient population. It does not appear to be as common as retinal dystrophies but was documented in eight patients (14.8%). Epilepsy (5.3%) and autism spectrum disorder (ASD) (5.3%) were also noted in three patients, respectively [55].

#### 4.2.5. Immune Symptoms

Although discussed in animal models in the literature, no studies in humans have been published to elucidate the possible immune manifestations apparent in subjects harboring *CACNA1F* pathogenic variants [18,19,20,21,22].

## 5. Discussion

The systematic literature review identified forty-eight patients diagnosed with *CACNA1A*-SHM1 and detailed the severe neurologic symptoms during and outside of attacks, capturing a comprehensive description of the phenotype and matching the case in all but two symptoms: phonophobia and seizures outside of hemiplegic attacks (Table 5). The case patient is nonverbal, so assessment of phonophobia is not possible.

In 2000, the grave symptoms affiliated with SHM1 surprised investigators when they identified the first de novo case [31] after exclusively studying families with inherited *CACNA1A* variants associated with the familial form, FHM1. The phenotypes for SHM1 and FHM1 are strikingly different. The literature review has highlighted the clinical differences between patients with SHM1 versus the more recognized FHM1 by providing the most complete description of their phenotype during and outside attacks. In alignment with these findings, the case patient exhibited the hallmark severe symptoms seen by patients with SHM1 in the literature, including early dystonia, paroxysmal events, cerebellar atrophy, and GDD, while the attacks were significantly more critical with prolonged encephalopathy.

In conjunction with the *CACNA1F* phenotype described in the scoping literature review, the patient presented with intellectual disability and ASD, although being consistent with the *CACNA1A* phenotype as well. Present in the case family is CSNB2A, similar to most of the subjects included in the review. Currently, the patient has not been evaluated due to sensitivity to reagents used for diagnosis. Finally, the immune manifestations do not align with the SHM1 phenotype reported in the literature, bringing into question the impact of the *CACNA1F* variant on the case study’s phenotype, but without data, conclusions remain uncertain.

### 5.1. Limitations

Different study designs were used for the systematic review including case reviews, cross-sectional studies, and observational cohorts. Some studies conducted systematic patient assessments while others focused on a few specific clinical features potentially under-reporting certain symptoms. It was assumed that if a symptom was not reported, it was not present, which may erroneously underestimate the prevalence of clinical features. Longitudinal data was rarely available, and until this case study, the natural history of SHM1 was not documented. For the case report, medical records came from four different children’s hospitals, and seven years of testing introduced variations in equipment and staffing, which can lead to interpretational differences in diagnostic images.

A rigorous systematic literature review was conducted using PRISMA guidelines, including two reviewers and an arduous procedure employing three databases to capture all documented cases. However, articles may have been overlooked. The primary review created the largest sample to date of 48 patients with SHM1. All patients were systematically assessed for a comprehensive evaluation.

### 5.2. Clinical Implications

This case report with the included systematic literature review and scoping review has multiple clinical implications. It clearly highlights that hemiplegic attacks associated with SHM1 are serious, unpredictable, and require an immediate, accurate diagnosis. Patients with SHM1 display significant challenges beginning in early childhood with paroxysmal events and GDD. Over time, the continuous periodic assaults are associated with increasing brain atrophy. This complete picture may help clinicians establish an earlier diagnosis, avoiding unnecessary invasive tests to rule out other infectious agents and provide the best supportive care possible. However, SHM1 is still understudied and lacks a complete phenotypic presentation, demanding future investigations to uncover a more comprehensive natural history of the disorder and potentially expose triggers that initiate these periodic attacks. The demonstrated case of an individual treated without any preventative medications allows us to assess the distribution of hemiplegic migraine attacks over time and highlights no clear pattern. Both severity and time of recurrence are difficult to estimate.

### 5.3. Future Research Directions

This investigation identified multiple knowledge gaps in need of future research. Longitudinal natural history studies are required to assess *CACNA1A*-SHM1 and to develop an age-dependent phenotype. Most data populated in clinical ontologies for *CACNA1A* lack the comprehensive phenotype terminology associated with the more severe SHM1, which is not fully captured in OMIM. Likewise, the differences in hemiplegic migraine are not captured in Human Phenotype Ontology, and neither is there a disease classification. Larger studies are essential to explore the genotype–phenotype correlation between specific variants and phenotypes and should consider epigenetic factors during the investigation, specifically gene expression patterns for *CACNA1A*.

## 6. Conclusions

With an accompanying systematic literature review, this case report offers a unique perspective into a rare, understudied disorder. It is the first documented case of the natural history of SHM1 without preventative therapeutics and the first description of a single patient harboring two calcium channel pathogenic variants. The review focusing on *CACNA1A*-SHM1 has not been reported previously in the literature. This investigation identified the clinical features and increased severity of SHM1 compared to FHM1 and demonstrates the demand for improved therapeutics to treat this disorder. The information provided may help diagnose new patients earlier and identify clinical features of concern. In this case, the patient’s symptoms correlate with nearly all the severe symptoms seen in patients with SHM1 in the literature. However, it is difficult to relate the immune phenotype in the case to its corresponding genotype due to the lack of comprehensive data for *CACNA1F,* highlighting the difficulties in rare disease research to attribute symptoms to specific modifying genes.

## Figures and Tables

**Figure 1 genes-14-00400-f001:**
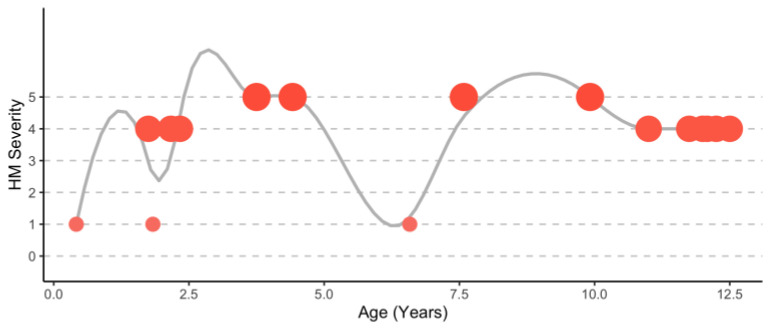
Graphic description of the natural history of the disorder.

**Figure 2 genes-14-00400-f002:**
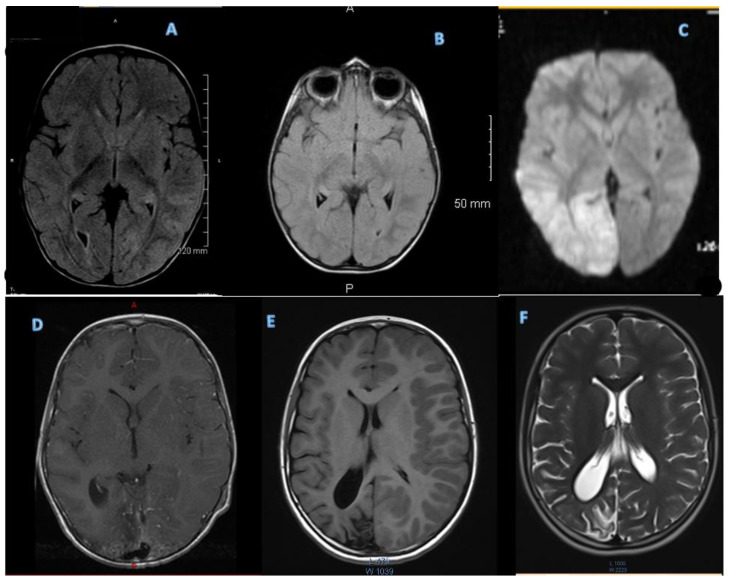
(**A**) Two years and 4 months (3rd attack—1st with hemiplegia) showed medial left temporal lobe edema (FLAIR imaging). (**B**) Two years and 5 months (no attack) normal (FLAIR imaging). (**C**) Diffusion-weighted imaging (DWI) at 3 years 9 months during 4th attack HM episode, showing increased signal in right occipital lobe. (**D**) Contrast-enhanced TI imaging at 4 years showing increased enhancement in right occipital lobe during 5th HM episode. (**E**) TI imaging at 7 years during 6th attack showing atrophy in right occipital lobe. (**F**) T2 imaging performed at 11 years during minor HM episode, demonstrating stable atrophy on the right occipital lobe without associated edema.

**Figure 3 genes-14-00400-f003:**
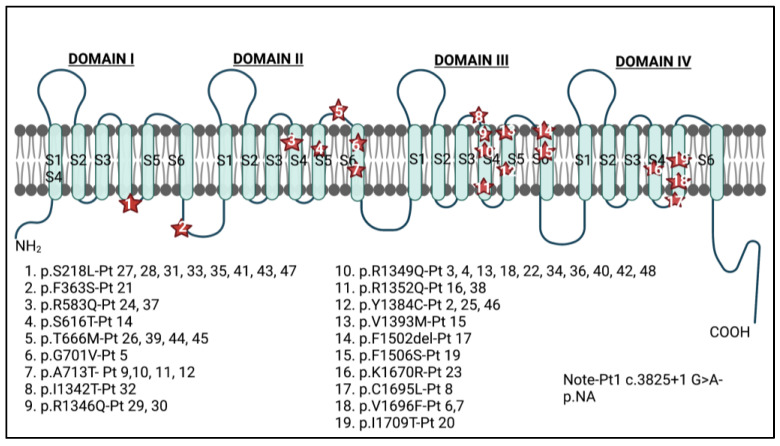
Schematic illustration of the Ca_v_2.1 alpha1A subunit with the de novo pathogenic variants associated with SHM1 identified in this systematic review. Variants were aligned to RefSeq transcript NM_001127221.1. The image shown is the full protein sequence including all exons. The variant in patient 18 was corrected due to a reporting error in the original paper. Created with BioRender.com.

**Table 1 genes-14-00400-t001:** Genes of the voltage-gated calcium channel superfamily.

Channel Type	Gene Name	Protein	Chromosome	Associated Condition-OMIM#	Primary Tissue Expression
L-type	*CACNA1S*	Ca_v_1.1	1q31-32	HOKPP1 ^1^-170400; MSH5 ^2^-601887; TTPP1 ^3^-188580	Muscle Skeletal
	*CACNA1C*	Ca_v_1.2	12p13.3	TS ^4^-601005; NEDHLSS ^5^-620029; LQT8 ^6^-618447; BRGDA3 ^7^-611875	GI, GU, Heart, Brain
	*CACNA1D*	Ca_v_1.3	3p14.3	SANDD ^8^-614896; PASNA ^9^-615474	Lung, Adrenal, Pituitary
	*CACNA1F*	Ca_v_1.4	Xp11.23	CORDX3 ^10^-300476; CSNB2A ^11^-300071; AIED ^12^-300600	Retina, GI, Lymph, Spleen
P/Q type	*CACNA1A*	Ca_v_2.1	19p13.1	EA2 ^13^-108500; SCA6 ^14^-183086; FHM1 ^15^-141500; DEE42 ^16^-617106	Brain-Cerebellum, Stomach
N-type	*CACNA1B*	Ca_v_2.2	9p34	KLEFS1 ^17^-610253; NEDNAH ^18^-618497; DYT23 ^19^-614860	Brain-Cerebellum, Cortex
R-type	*CACNA1E*	Ca_v_2.3	1q25031	DEE69 ^20^-618285	Brain (throughout)
T-type	*CACNA1G*	Ca_v_3.1	17q22	SCA42 ^21^-616795; SCA42ND ^22^-618087	Brain, Endometrium
	*CACNA1H*	Ca_v_3.2	16p13.3	HALD4 ^23^-617027; ECA6 ^24^-611942 [Disputed gene-disease relationship]	GU, GI
	*CACNA1I*	Ca_v_3.3	22q13	NEDSIS ^25^-620114	Brain, Thyroid

^1^ Hypokalemic periodic paralysis, type 1. ^2^ Malignant hyperthermia, susceptibility to, 5. ^3^ Thyrotoxic periodic paralysis, susceptibility to, 1. ^4^ Timothy syndrome. ^5^ Neurodevelopmental disorder with hypotonia, language delay, and skeletal defects with or without seizures. ^6^ Long QT syndrome 8. ^7^ Brugada syndrome 3. ^8^ Sinatrial node dysfunction and deafness. ^9^ Primary aldosteronism, seizures, and neurologic abnormalities. ^10^ Cone–rod dystrophy, X-linked, 3. ^11^ Night blindness, congenital stationary, type 2A. ^12^ Aland island eye disease. ^13^ Episodic ataxia, type 2. ^14^ Spinocerebellar ataxia 6. ^15^ Migraine, familial hemiplegic, 1. ^16^ Developmental and epileptic encephalopathy 42. ^17^ Kleefstra syndrome 1. ^18^ Neurodevelopmental disorder with seizures and nonepileptic hyperkinetic movements. ^19^ Dystonia 23. ^20^ Developmental and epileptic encephalopathy 69. ^21^ Spinocerebellar ataxia 42. ^22^ Spinocerebellar ataxia 42, early onset, severe, with neurodevelopmental deficits. ^23^ Hyperaldosteronism, familial, type IV. ^24^ Epilepsy, childhood absence, susceptibility to, 6 (This association is disputed; see ClinGen review at https://search.clinicalgenome.org/kb/gene-validity/CGGV:assertion_c3208279-a38d-4b38-b4f9-3af2610c4936-2018-07-30T040000.000Z accessed on 8 December 2022). ^25^ Neurodevelopmental disorder with speech impairment and with or without seizures.

**Table 2 genes-14-00400-t002:** Description of each attack in chronological order.

Possible triggers	High alt.								X								
	Airborne allergen							X		X	X	X	X	X	X	X	X
	Head trauma						X										
	Vitamins					X											
	Food-diet	X	X	X	X			X									
Symptoms	Abn. MRI		X			X	X	X		X							
	Abn. EEG		X			X	X	X		X							
	Abn. Labs		X		X	X	X	X	X	X							
	Transient blindness							X		X	X	X		X		X	
	Encephalopathy					X	X	X		X	X	X	X	X	X		
	Hemiplegia					X	X	X	X	X	X	X		X		X	X
	Fever		X		X	X	X	X		X	X	X		X		X	X
	Status Epilepticus						X	X		X							
	Seizure		X		X		X	X		X							
	Tachycardia		X			X				X			X	X	X	X	X
	Apnea		X		X			X									
	Dyskinesia												X	X	X	X	X
	Coma		X		X	X	X	X		X							
	Lethargy	X	X	X	X	X	X	X	X	X	X	X					
	Vomiting	X	X	X	X	X	X	X	X	X	X	X					
	Duration	<20 s	5 days	<24 h	4 days	5 days	5 days	11 days	<6 h	16 days	6 days	4 days	3 days	5 days	5 days	3 days	2 days
	Age	5 M	1 Y 9 M	1 Y 10 M	2 Y 2 M	2 Y 4 M	3 Y 9 M	4 Y 5 M	6 Y 7 M	7 Y 7 M	9 Y 11 M	11 Y	11 Y 9 M	12 Y	12 Y 1 M	12 Y 3 M	12 Y 6 M
	Episode	1	2	3	4	5	6	7	8	9	10	11	12	13	14	15	16

**Table 3 genes-14-00400-t003:** Reported symptoms during hemiplegic migraine attacks [3,4,5,6,7,8,10,12,23,24,25,27,28,29,30,31,32,33,34,35,36,37,38,39,40,41].

Symptoms During Hemiplegic Attacks	N (%)
Migraine with aura	33/33 (100%)
Hemiplegia	33/33 (100%)
Coma/altered consciousness	25/33 (75.8%)
Seizure (including status epilepticus)	25/33 (75.8%)
Cerebral edema	15/33 (45.5%)
EEG slowing or spike waves	12/33 (36.4%)
Eye abnormalities (including transient blindness)	11/33 (33.3%)
Encephalopathy	11/33 (33.3%)
Dystonic storms	4/33 (12.1%)
Lethargy	3/33 (9.1%)
Fever (noninfectious)	17/33 (51.5%)
Vomiting/nausea	13/33 (39.4%)
Apnea	4/33 (12.1%)
Other symptoms	
Speech difficulties	7/33 (21.2%)
Photophobia	4/33 (12.1%)
Phonophobia	2/33 (6.1%)
Abnormal CSF (including high IL-6)	2/33 (6.1%)

**Table 4 genes-14-00400-t004:** Reported symptoms outside of hemiplegic migraine attacks [3,4,5,6,7,8,10,12,23,24,25,27,28,29,30,31,32,33,34,35,36,37,38,39,40,41].

Symptoms Outside of Hemiplegic Attacks	No. (%)
Movement disturbance	37/46 (80.4%)
Ataxia	35/46 (76.1%)
Dystonia	29/46 (63.0%)
Hypotonia	24/46 (52.2%)
Myoclonus	6/46 (13.0%)
Pyramidal signs	6/46 (13.0%)
Tremor	11/46 (23.9%)
Dyskinesia	4/46 (8.7%)
Global developmental delay	29/46 (63.0%)
Speech disturbance	13/46 (28.3%)
Intellectual disability	30/46 (65.2%)
Severe-IQ (20–35)	7/30 (23.4%)
Moderate-IQ (36–49)	4/30 (13.3%)
Mild-IQ (50–69)	9/30 (30.0%)
Undefined	10/30 (33.3%)
Brain atrophy	28/37 (75.7%)
Progressive	9/28 (32.1%)
Paroxysmal actions	9/46 (19.6%)
Abnormal eye movements	22/46 (47.8%)
Seizures	11/46 (23.9%)

**Table 5 genes-14-00400-t005:** The case study patient’s symmetry to the *CACNA1A*-SHM1 phenotype. (* notes common to FHM1 as well but less common were coma, cerebral edema, tremor, and seizures).

Clinical Features	Case Study	SHMI (*CACNA1A)*
**Symptoms During Hemiplegic Attacks**		
Migraine with aura	+	+ *
Hemiplegia	+	+ *
Neurologic manifestations		
Coma/altered consciousness	+	+ *
Seizure (including status epilepticus)	+	+
Cerebral edema	+	+ *
EEG slowing or spike waves	+	+
Eye abnormalities	+	+
Encephalopathy	+	+
Dystonic storms	+	+
Lethargy	+	+ *
Autonomic manifestations		
Fever (noninfectious)	+	+ *
Vomiting/nausea	+	+
Apnea	+	+
Other Symptoms		
Speech difficulties	+	+ *
Photophobia	+	+
Phonophobia		+
Abnormal CSF	+	+
** Symptoms Outside of Hemiplegic Attacks **		
Movement disorders		
Ataxia	+	+
Dystonia	+	+
Pyramidal signs	+	+
Tremor	+	+ *
Dyskinesia	+	+
Global developmental delay	+	+
Verbal conditions	+	+
Intellectual disability	+	+
Brain atrophy	+	+ *
Paroxysmal actions	+	+
Abnormal eye movements	+	+ *
Seizures		+ *

## Data Availability

The data presented in this study are available in the Appendix A.

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
