# Peer review of "Concomitant Calcium Channelopathies Involving CACNA1A and CACNA1F: A Case Report and Review of the Literature"

_genes, 2023, doi:10.3390/genes14020400_

Round 1

Reviewer 1 Report

This is a well written case report an literature review on a calcium channelopathy.  I find it interesting and informative, but there is some room for improvement. 

Points to address:

There is a strg emphasis on CACNA1A for good reasons, but I find that the CACNA1F component fell by the wayside - the reader is set up to expect a discussion of the CACNA1F related symptoms by very little is presented.  This critical paper by Bech-Hansen et al Nature Genetics is not cited, and compared to the in depth discussion on CACNA1A, this component falls short.  I am not expecting something as deep as the CSCNA1A section, but ate least talk in more depth about the many mutations in Cav1.4 that have been identified, and cite some original papers, and then relate this back to the family history. 

Section 1.3 should really come first before current section1;1 and 1.2

References 12 and 21 are duplicates of each other

Table on primary expression for Cav1.4 the retina should be added as well

Reviewer 2 Report

The current paper, which was submitted to Genes, presents a case study of a child who carries concomitant rare two calcium channel pathogenic mutations involving CACNA1A and CACNA1F. Alterations in calcium channels lead to channelopathies, primarily manifested in the central nervous system.  A thorough literature review on the Sporadic Hemiplegic Migraine type 1 syndrome is also conducted.

Overall, the case is well-presented in minute detail and appropriately contrasted to the literature. Please see my remarks below:

-In the described case, the CACNA1F mutation has been postulated to be the origin of immunological manifestations such as elevated IFN-g. However, it is preferable if the authors stress this variant's possible relevance in the discussion and compare it to further data from the literature, if available.

-What may explain the lack of coma, lethargy, and vomiting in episodes 12-16 (table 2), as well as the emergence of dyskinesia?
